# Preparation of Multicolor Photoluminescent Carbon Dots by Tuning Surface States

**DOI:** 10.3390/nano9040529

**Published:** 2019-04-03

**Authors:** Kai Jiang, Xiangyu Feng, Xiaolu Gao, Yuhui Wang, Congzhong Cai, Zhongjun Li, Hengwei Lin

**Affiliations:** 1State Key Laboratory of Coal Mine Disaster Dynamics and Control, Department of Applied Physics, Chongqing University, Chongqing 400044, China; jiangkai@nimte.ac.cn (K.J.); czcai@cqu.edu.cn (C.C.); 2Key Laboratory of Graphene Technologies and Applications of Zhejiang Province & Ningbo Institute of Materials Technology & Engineering (NIMTE), Chinese Academy of Sciences (CAS), Ningbo 315201, China; fengxiangyu@nimte.ac.cn (X.F.); gaoxiaolu@nimte.ac.cn (X.G.); wangyuhui@nimte.ac.cn (Y.W.); 3College of Chemistry and Molecular Engineering, Zhengzhou University, Zhengzhou 450001, China; lizhongjun@zzu.edu.cn

**Keywords:** carbon dots, multicolor photoluminescence, surface state, light-emitting diodes

## Abstract

The achievements of multicolor photoluminescent (PL)-emissive carbon dots (CDs), particularly red to near infrared (NIR), are critical for their applications in optoelectronic devices and bioimaging, but it still faces great challenges to date. In this study, PL emission red-shifts were observed when tartaric acid (TA) was added into m-phenylenediamine (mPD) or o-phenylenediamine (oPD) solutions as carbon sources to prepare CDs, i.e., from blue to green for mPD and from yellow-green to red for oPD. Morphology and structure analyses revealed that the increased surface oxidation and carboxylation were responsible for the red-shifts of emission, indicating that TA played a key role in tuning the surface state of CDs. These factors could be employed as effective strategies to adjust PL emissions of CDs. Consequently, multicolor PL CDs (i.e., blue-, green-, yellow-green- and red-emissive CDs) can be facilely prepared using mPD and oPD in the absence and presence of TA. Particularly, the obtained red-emissive CDs showed a high PL quantum yield up to 22.0% and an emission covering red to NIR regions, demonstrating great potentials in optoelectronic devices and bioimaging. Moreover, multicolor phosphors were further prepared by mixing corresponding CDs with polyvinylpyrrolidone (PVP), among which the blue, green, and red ones could serve as three primary color phosphors for fabricating multicolor and white light-emitting diodes (LEDs). The white LED was measured to show a Commission Internationale de L’Eclairage (CIE) 1931 chromaticity coordinate of (0.34, 0.32), a high color rendering index (CRI) of 89, and a correlated color temperature (CCT) of 5850 K, representing one of the best performances of white LEDs based on CDs.

## 1. Introduction

Since the traditional photoluminescent (PL) materials (e.g., semiconductor quantum dots, organic fluorescent dyes, and rare earth-based phosphors) exhibit drawbacks of quick photobleaching, complicated preparation processes, severe toxicity, and/or detrimental effects to humans and the environment, the exploitation of new kinds of PL materials are still highly desirable [1,2,3]. In recent years, carbon dots (CDs) have emerged to be an attractive type of PL material because of their numerous merits, such as tunable emissions, high photostabilities, excellent biocompatibilities, cost-effective preparations, and easy surface functionalization [4,5,6,7]. Benefited from these superior properties, CDs are deemed to be one of the most promising alternatives to conventional PL materials [8,9], and their potential applications, including bioimaging, theranostics, sensing, photocatalysis, and light-emitting diodes (LEDs), have been widely explored [10,11,12,13,14,15,16,17].

The difficulties in preparing multicolor emissive CDs, particularly red to near-infrared (NIR) with superior PL properties (e.g., high quantum yield (QY) and color purity) and easily controllable processes is one of the major obstacles in driving their applications from the laboratory to the market [8,9]. Although numerous methods for the modulation of PL colors of CDs have been reported recently through surface modification [18,19,20], solvent engineering [21,22,23,24], conjugated carbon precursor introduction [25,26,27], heteroatom doping [28,29,30], and surface defect regulation [31,32,33,34], these approaches usually demanded elaborate control of the reaction conditions and/or tedious separation/purification procedures. Thus, most of these methods might be difficult for reproduction and not suitable for large-scale preparation as well. Due to the superior tissue penetration depth and one of the critical building blocks for fabricating full-color displays and white-lighting devices [23,35,36,37], the achievement of CDs with superior red to NIR emissions is indispensable, but this is still a great challenge [23,26,35,36,37,38]. Therefore, the exploitation of reproducible, controllable, and scalable strategies to prepare multicolor emissive CDs is highly significant for boosting their applications in the future.

The PL mechanism of CDs is a longstanding debate, but emission from carbon cores, surface defects, and molecular states are some of the mostly acceptable origins [39]. Consequently, the emission features of CDs could be regulated by controlling their size (mainly referring to sp2 carbon domains), heteroatom doping, and surface functionalization [7,9,40,41,42]. More impressively, despite various structure motifs that are usually simultaneously in existence in CDs and responsible for their PL emission, the major contributions from certain key factors has been demonstrated. For instance, Fan et al. reported multicolor band-gap-based CDs with PL emissions covering from blue to red by increasing the particle sizes [43,44]. Xiong et al. and Sun et al., respectively, obtained a series of CDs with tunable PL colors and proved that the emission was mainly controlled by the surface state [34,45]. In addition, several studies have shown that nitrogen doping could trigger PL red-shifts of CDs [46,47,48,49], and Zbořil et al. further clarified that graphitic nitrogen doping should be the major reason to play such a role [50].

In previous work we had prepared red, green, and blue emissive CDs using p-, o-, or m-phenylenediamines (abbreviated as pPD, oPD, and mPD, respectively) as starting materials [25]. These CD emissions, however, are not the ideal three primary colors (i.e., green and red emissive CDs inclined to yellow-green and orange regions, respectively). In this study, we attempted to further optimize emission wavelengths of these synthetic systems through adjusting their structures and surface states. Impressively, when tartaric acid (TA) was added to oPD or mPD solutions and then subjected to a solvothermal reaction, obvious emission red-shifts were observed from the prepared CDs compared to those without the addition of TA (i.e., from blue to green for mPD and yellow-green to red for oPD). Morphology and structure/composition analyses revealed that the increased surface oxidation and carboxylation should be responsible for the observed red-shifts of emission. Therefore, such factors could be employed as effective strategies to adjust the PL emission wavelength of CDs. Based on these findings, multicolor PL CDs (i.e., blue, green, yellow-green, and red emissive CDs, abbreviated as b-CDs, g-CDs, y-CDs, and r-CDs, respectively) were facilely prepared using mPD or oPD in the absence and presence of TA. Particularly, the obtained r-CDs exhibited a respectable PL QY of 22.0% and an emission covering red to NIR regions, indicating promising applications in bioimaging and building optoelectronic devices. Moreover, multicolor phosphors (blue, green, yellow-green, and red) can be further obtained by simply mixing these CDs with polyvinylpyrrolidone (PVP), among which the blue, green, and red ones were shown to be the ideal three primary color phosphors for fabricating multicolor and white light-emitting diodes (LEDs). The white LED was measured to show a Commission Internationale de L’Eclairage (CIE) 1931 color coordinate to be (0.34, 0.32), which is very close to the pure white-lighting color coordinate of (0.33, 0.33). Additionally, this white LED also displayed a high color rendering index (CRI) of 89 and relatively low correlated color temperature (CCT) of 5850 K, representing one of the best performances of white LEDs based on CDs.

## 2. Materials and Methods

### 2.1. Reagents

Reagent grades of o-, m-, and p-phenylenediamines (oPD, mPD, and pPD) and tartaric acid (TA) were purchased from Aladdin (Shanghai, China). Ethanol, methanol, acetonitrile, methylene chloride, and polyvinyl pyrrolidone (PVP) (K-30) were provided by Sinopharm Chemical Reagent Co., Ltd. (Shanghai, China). All chemicals were used as received without further purification unless otherwise specified. Deionized (DI) water was used throughout this study.

### 2.2. Synthesis of Multicolor Emissive Carbon Dots (CDs)

The b-CDs and y-CDs were prepared according to our previous study [25]. The g-CDs and r-CDs were prepared using the same procedures as that of the b-CDs and y-CDs, but with the addition of TA. Typically, mPD (or oPD) (0.90 g) and TA (1.2 g) were dissolved in ethanol (45 mL), and then the solution was transferred into a poly(tetrafluoroethylene)-lined autoclave. After heating at 180 °C in an oven for 12 h and cooling down to room temperature naturally, a gray (or reddish brown) suspension was obtained. The crude products were then purified with silica column chromatography using a mixture of methylene chloride and methanol as eluent. After removing solvents and further drying in a vacuum oven, the g-CDs and r-CDs were finally obtained in 30~40 wt% yields.

### 2.3. Procedure

Preparation of multicolor emissive phosphors: Multicolor emissive phosphors were prepared referring to a previously described method [26]. Typically, the b-CDs (g-CDs, y-CDs, or r-CDs) solution (5.0 mg/mL in ethanol, 0.5 mL) was mixed with PVP solution (1.2 g/mL in ethanol, 1.5 mL) under ultrasonic for 0.5 h. Then, the mixed solution was dropped on a clean glass substrate and dried in a vacuum oven at 60 °C for 24 h. The dried blocks were thoroughly ground in agate mortar to obtain fine phosphor powders.

Fabrication of multicolor LEDs: A UV-LED chip with the peak emission wavelength centered at 365 nm was used for the fabrication of multicolor LEDs. Typically, 0.2 g phosphor was mixed with 2.0 g epoxy resin (i.e., epoxy resin A:epoxy resin B = 3:1, w/w) thoroughly, and the obtained phosphor-epoxy mixture was coated on the surface of the LED chip to obtain multicolor LEDs.

Fabrication of white LED: A UV-LED chip with the peak emission wavelength centered at 365 nm was used for the fabrication of a white LED. A certain amount of b-CDs, g-CDs, and r-CDs phosphors (i.e., b-CDs@PVP:g-CDs@PVP:r-CDs@PVP = 2:5:8, w/w/w) was mixed with silicone thoroughly, and the obtained phosphor-silicone mixture was coated on the surface of the UV-LED chip to produce white LED.

Measurement of quantum yields (QYs): The absolute quantum yields (Φ) of the CDs and phosphors were measured on a QE-2100 quantum efficiency measurement system (Japan Otsuka Electronics, Tokyo, Japan), and the results were summarized in the Appendix A, respectively. To obtain more reliable results, five replicate measurements and rhodamine 6G (Φ = 0.95) as a reference were performed.

### 2.4. Equipment and Characterization

Transmission electron microscopy (TEM) observations were performed on a Tecnai F20 microscope. Atomic force microscope (AFM) measurements were carried out with Veeco Dimension 3100V. X-ray photoelectron spectroscopy (XPS) was carried out with ESCALAB 250Xi (Thermo Scientific, Waltham, MA, USA). Fourier transform infrared (FT-IR) spectra were obtained on a Nicolet 6700 FT-IR spectrometer (Thermo Nicolet Corp., Madison, WI, USA). Photoluminescence emission and excitation spectra were measured on a Hitachi F-4600 spectrophotometer (Hitachi, Tokyo, Japan) equipped with a Xe lamp at ambient conditions. UV-Vis absorption spectra were recorded on a PERSEE T10CS UV-Vis spectrophotometer (Persee, Beijing, China). PL lifetime was measured using Fluorolog 3-11 (HORIBA Jobin Yvon, Kyoto, Japan). PL Quantum yields were measured on a QE-2100 quantum efficiency measurement system (Otsuka Electronics, Japan). Photographs were taken using a Canon camera (EOS 550, Tokyo, Japan) under excitation by a hand-held UV lamp (365 nm).

## 3. Results and Discussion

### 3.1. Preparation and Optical Properties of CDs

The b-CDs, g-CDs, y-CDs, and r-CDs were facilely prepared using mPD (or oPD) in the absence and presence of TA as carbon precursors via a solvothermal method, and they were purified by column chromatography (refer to the Experimental section for details) (Figure 1a). These purified CDs were dispersed in common solvents and showed transparency with different colors (e.g., Figure 1b in ethanol). Under irradiation by UV light (i.e., 365 nm), these CD solutions displayed bright blue, green, yellow-green, and deep-red emissions (Figure 1c). PL spectra measurements showed that the b-CDs, g-CDs, and y-CDs were single-peak-dominated emissions with maxima located at 435, 510, and 535 nm, respectively. The r-CDs, however, exhibited a multi-peak emission (i.e., λ_em_(max) = 608, 650, and 714 nm) covering the region from red to NIR (Figure 1d). These PL spectra were in good accordance with their emission colors.

UV-Vis absorptions, PL emissions, and excitation properties of the b-CDs, g-CDs, y-CDs and r-CDs were investigated and compared to elucidate the effects of adding TA. As shown in Figure 2, both of the g-CDs and r-CDs (prepared from mPD or oPD with the addition of TA) exhibited apparent absorption and emission red-shifts in comparison to the b-CDs and y-CDs (prepared from mPD or oPD without the addition of TA). Specifically, the UV-Vis absorption maxima red-shifted from 247 to 252 nm and 355 to 460 nm when comparing b-CDs and g-CDs, and from 260 to 288 nm and 426 to ~540 nm when comparing y-CDs and r-CDs. The observed short- and long-wavelength absorption bands here were generally attributed to the π→π* and n→π* transitions of CDs, respectively, in reference to the literature [34,45]. In addition, both of the g-CDs and r-CDs showed excitation wavelength-independent PL features (Appendix A), indicating only one emission transition that dominated their PL. The PL red-shifts from 435 nm (b-CDs, from mPD only) to 510 nm (g-CDs, from mPD with the addition of TA) and from 535 nm (y-CDs, from oPD only) to a multi-peak red to NIR band (r-CDs, from oPD with the addition of TA) were also observed. These significant optical property variations implied remarkable differences in chemical structures of CDs that were prepared from mPD or oPD in the absence and presence of TA. In addition, the PL excitation spectra of these CDs were found to be similar to their UV-Vis absorption spectra (Appendix A), demonstrating that their emissions should be correlated to their absorption-relevant moieties.

Subsequently, PL QYs of the g-CDs and r-CDs were measured and determined to be 28.22% and 22.0%, respectively (Appendix A), which were higher than that of the b-CDs and y-CDs (i.e., 4.8% and 10.4%, respectively) [25]. These results indicated that the addition of TA not only changed the emission wavelength, but also increased QYs for the corresponding CDs. Based on the time-resolved PL spectra and corresponding fittings, both of the g-CDs and r-CDs were found to have mono-exponential decays with lifetimes of 4.86 ns and 2.18 ns, respectively (Appendix A), implying that only one radiation transition dominated their emissions. Note that unlike other CDs, the r-CDs displayed a multi-peak emission feature (i.e., λ_em_(max) = 608, 650, and 714 nm), and the closely similar decay behaviors and fitted parameters at these emission wavelengths indicated that the multi-peak emission resulted from the same origin (Appendix A) [51].

### 3.2. Morphologies and Structure Analyses of CDs

To obtain insights of the emission red-shifts caused by the addition of TA, morphology and structure analyses of g-CDs and r-CDs were carried out and compared with b-CDs and y-CDs. First, the morphologies of g-CDs and r-CDs were investigated by transmission electron microscope (TEM) and atomic force microscopy (AFM). As shown in Figure 3a,e, both of the g-CDs and r-CDs were well-dispersed and exhibited quasi-spherical particle shapes with amorphous features, which were further confirmed by high-resolution TEM (inserts of Figure 3a,e). The AFM images showed that the heights were approximately 1–2 nm for the g-CDs (Figure 3c,d) and 3–4 nm for the r-CDs (Figure 3g,h). Importantly, the size distribution showed the average diameters of g-CDs and r-CDs were 3.6 and 4.8 nm, respectively (Figure 3b,f), which were smaller than that of the b-CDs (6.0 nm) and y-CDs (8.2 nm) [25], demonstrating that particle size was not the major factor to cause the red-shifts [31,43,44].

Subsequently, the chemical composition and surface functional groups of these CDs were examined using Fourier transform infrared (FT-IR) and X-ray photoelectron spectroscopy (XPS). As shown in Figure 4a,b, the characteristic FT-IR peaks emerged at 1732 cm^−1^ and 3426 (or 3429) cm^−1^ for the g-CDs and r-CDs in comparison with the b-CDs and y-CDs, which were attributed to C=O and –OH stretching vibrations [34,37,48,52], respectively. The XPS measurements exhibited that all of these CDs consisted of C, N, and O elements (Appendix A). From their relative contents, the atomic ratios between O and C increased from b-CDs (0.14) to g-CDs (0.27) and from y-CDs (0.21) to r-CDs (0.41) (Appendix A), confirming an obvious increase in oxidation degree and/or introduction of oxygen-containing functional groups when TA was added for the preparation of CDs by mPD or oPD. In the high resolution XPS of the g-CDs and r-CDs, their C 1s spectra were deconvoluted into four Gaussian peaks corresponding to C=C (284.6 eV), C–N (285.4–285.5 eV), C–O (286.3–286.5 eV), and O=C–OH (288.3–288.7 eV) (Figure 4c,d) [34,37,48,52]. The deconvoluted results of N 1s spectra showed three peaks located at 398.2–398.4 eV, 399.2–399.4 eV, and 400.1 eV, assigned to pyridinic N, amino N, and pyrrolic N, respectively (Appendix A) [45,53]. The O 1s spectra were fitted with two components of C=O (531.7 eV) and C–O (532.8 eV) (Appendix A). These results were in good accordance with the observed signals of 3216–3261 cm^−1^, 1622–1633 cm^−1^, 1475–1495 cm^−1^, and 1024–1134 cm^−1^ in the FT-IR spectra, which were attributed to N–H, C=C/C=N, C–N=, and C–O/C–N vibrations, respectively [37,45,48,52,54]. Remarkably, significant increases of C–O (286.3–286.5 eV) and O=C–OH (288.3–288.7 eV) components from b-CDs to g-CDs and from y-CDs to r-CDs were clearly observed based on the C 1s XPS fitting results (Figure 4c,d) [25]. More specifically, the relative contents of C–O bonds increased from 3.37% (b-CDs) to 26.66% (g-CDs) and from 6.03% (y-CDs) to 34.96% (r-CDs); meanwhile, the carboxyl group (O=C–OH) appeared and accounted for 7.34% and 10.37% in g-CDs and r-CDs, respectively (Table 1). These findings demonstrated that the oxidation degree and carboxylation were obviously enhanced when TA was added for the preparation of CDs by mPD or oPD, which could be important factors to induce red-shifts in PL emission of CDs.

### 3.3. Photoluminescent (PL) Mechanism of CDs

Due to the complicated and uncertain chemical structures, PL mechanisms of CDs is still under debate. Nevertheless, two possible emission origins are tentatively accepted by the research community. The first one is a band-gap-based carbon core state that contains conjugated π-domains, and the other is a defect and molecular fluorophore-relevant surface state [39]. As to the former mechanism, the particle sizes of CDs, or more precisely the size of the contained sp2 domains, would influence their PL positions [31,43,44]. In our case here, however, the size of CDs did not play an important role in their emission wavelengths based on the morphology characterizations (TEM and AFM). From composition and structure examinations, the increased contents of C=O and carboxyl groups were found to be the most different between CDs that were prepared in the absence and presence of TA. It is known that the surface defects were primarily created by surface oxidation, which could behave as capture centers for excitations and induce PL emissions [33,39,45]. Consequently, with the higher degree of surface oxidation a higher emission efficiency should be obtained. This deduction may explain the observed higher QYs of the g-CDs and r-CDs than the b-CDs and y-CDs, respectively. In addition, it was reported that the carboxyl groups that coupled on the sp2 carbon framework could cause obvious local distortions and, thus, narrow their energy gaps [34,37,48]. Based on such knowledge, the emerged surface carboxyl groups on the g-CDs and r-CDs were considered to be responsible for their emission red-shifts compared with the b-CDs and y-CDs, respectively. It is worthy to note that the PL properties of the prepared CDs in this study were similar to molecular fluorophores, so their emissions should rise from their surfaces that contained conjugated carbon frameworks coupled with oxygen functional groups. The band gaps of these chemical structures were strongly dependent on the degree of oxidation and carboxylation.

### 3.4. Applications of CDs in Multicolor and White LEDs

Due to the superior photophysical properties of these CDs (e.g., tunable PL emissions and high QYs), their potential applications in fabricating multicolor and white LEDs were preliminarily investigated. To prevent possible aggregation-induced quenching, multicolor phosphors were firstly prepared by simply dispersing these CDs in highly transparent polyvinylpyrrolidone (PVP) (please refer to the Experimental section for details). As shown in Appendix A, bright blue, green, yellow-green, and red emissions can be observed from the corresponding CDs@PVP phosphors under UV light irradiation (365 nm). The PL emission behaviors of these CD-based phosphors were similar to that in solution (Appendix A), indicating negligible interactions between the CDs and PVP. Moreover, the CD-based phosphors also remained relatively high QYs (i.e., 8.82%, 14.44%, 16.13%, and 14.81% for b-CDs@PVP, g-CDs@PVP, y-CDs@PVP, and r-CDs@PVP, respectively, Appendix A). Subsequently, multicolor LEDs were fabricated by encapsulating the corresponding CDs@PVP phosphors on a commercialized LED chip (365 nm). Their lighting images are offered in Figure 5a, that is the monochromatic blue, green, yellow-green, and red LEDs that were able to be obtained. Based on these LED emission spectra (Appendix A), the CIE coordinates of these LED emissions were determined, among which the blue, green, and red ones displayed a triangular distribution (Figure 5b). According to the Glassman color mixing law [25,55,56], the three primary color emissions located at the vertices of the triangle can be employed to achieve full-color tuning, demonstrating great potentials of these CD-based phosphors for multicolor lighting and full-color display devices.

To fabricate a white LED, a 365 nm UV-LED chip was selected as an excitation source. Firstly, the b-CDs@PVP, g-CDs@PVP, and r-CDs@PVP phosphors were uniformly mixed with epoxy resin A and B, and then the mixture was deposited on the UV-LED chip. Through adjusting and optimizing the ratios of the three phosphors (e.g., b-CDs@PVP:g-CDs@PVP:r-CDs@PVP = 2:5:8, w/w/w), an excellent white LED was obtained with a CIE 1931 coordinate of (0.34, 0.32), which was very close to the pure white light color coordinate of (0.33, 0.33). The emission spectrum of the white LED consisted of three emission bands centered at 435, 510, and 640 nm (Figure 5c), which were assigned to the emissions of b-CDs@PVP, g-CDs@PVP, and r-CDs@PVP, respectively. Moreover, the CCT and CRI of the white LED were also measured and determined to be 5850 K and 89, respectively, representing superior white light emission performances. These results demonstrate potential applications of the as-prepared CDs for the next generation of lighting devices.

## 4. Conclusions

In summary, a controllable and effective method has been developed for the preparation of multicolor PL CDs (i.e., blue, green, yellow-green, and red) by taking mPD or oPD in the absence and presence of TA as carbon sources in this study. The addition of TA was found to increase the degree of surface oxidation and carboxyl groups on prepared CDs, which were further evidenced to play a key role in the resulting PL emission red-shifts (i.e., from blue to green for mPD and from yellow-green to red for oPD in the absence and presence of TA, respectively). Moreover, these PL-tunable CDs can be further utilized as phosphors, fabricated multicolors, and white LEDs, demonstrating their potential applications in full-color displays and white lighting devices. The fabricated white LED was measured to show a close, pure white CIE chromaticity coordinate of (0.34, 0.32), a high CRI of 89, and a relatively low CCT of 5850 K, representing one of the best performances of CD-based white LEDs. This work is regarded as a primary step in realizing a rational design of CDs with specific optical properties. Thus, this work will help to boost their practical applications in the future.

## Figures and Tables

**Figure 1 nanomaterials-09-00529-f001:**
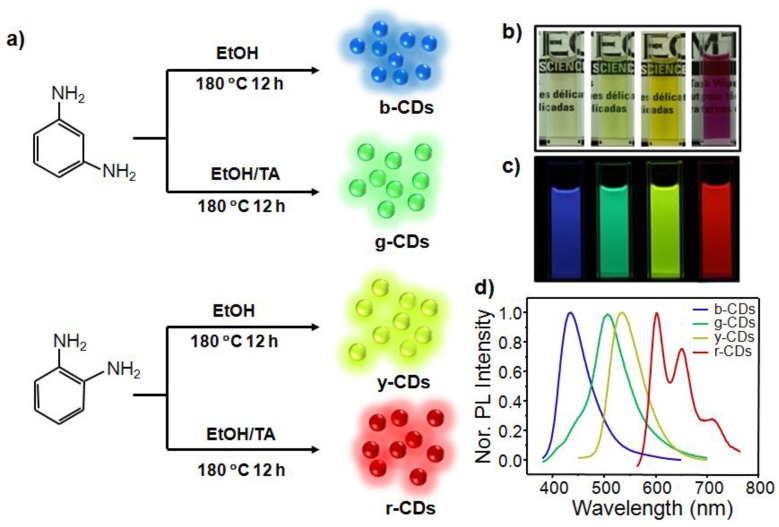
(**a**) Schematic illustration of the preparation process for multicolor carbon dots (CDs) using m- and o-phenylenediamines in the absence and presence of tartaric acid (TA), respectively; (**b**,**c**) Photographs of the obtained b-CDs, g-CDs, y-CDs, and r-CDs dispersions in ethanol under daylight (**b**) and 365 nm UV light (**c**), respectively; (**d**) Normalized photoluminescent (PL) emission spectra of the multicolor CDs (in ethanol) under excitation of 365 nm.

**Figure 2 nanomaterials-09-00529-f002:**
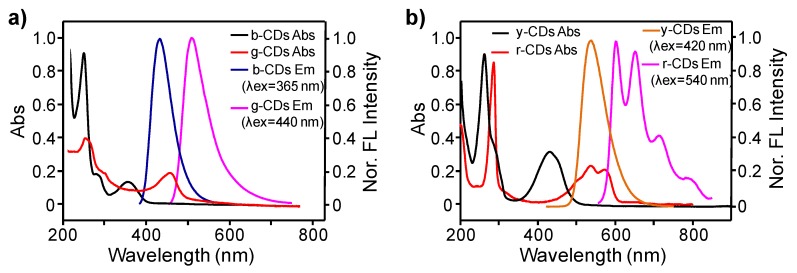
(**a**) Comparison of the UV-Vis absorption and PL emission spectra of b-CDs and g-CDs; (**b**) Comparison of the UV-Vis absorption and PL emission spectra of y-CDs and r-CDs.

**Figure 3 nanomaterials-09-00529-f003:**
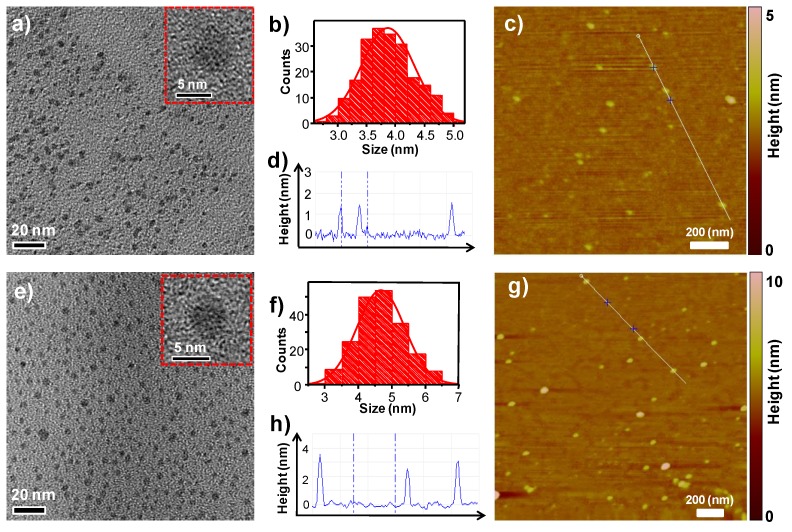
(**a**) TEM and high resolution TEM images of the g-CDs; (**b**) Size distribution of the g-CDs based on the TEM; (**c**) AFM image of the g-CDs; (**d**) height-profile analysis of the g-CDs along the line in (c); (**e**) TEM and high resolution TEM images of the r-CDs; (**f**) Size distribution of the r-CDs based on the TEM; (**g**) AFM image of the r-CDs; (**h**) height-profile analysis of the r-CDs along the line in (g).

**Figure 4 nanomaterials-09-00529-f004:**
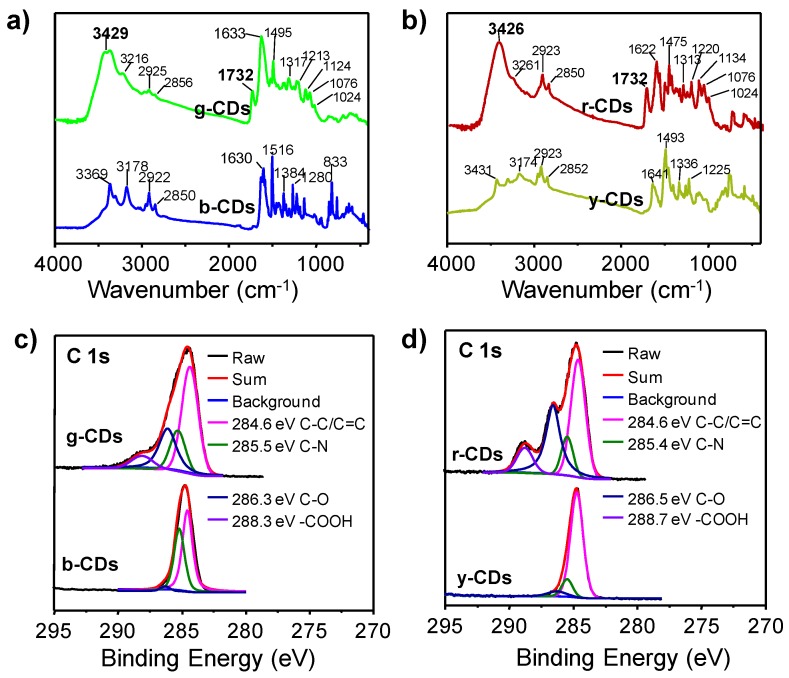
(**a**) Comparison of Fourier transform infrared (FT-IR) spectra of the b-CDs and g-CDs; (**b**) Comparison of FT-IR spectra of the y-CDs and r-CDs; (**c**) Comparison of high resolution of C 1s X-ray photoelectron spectroscopy (XPS) spectra of the b-CDs and g-CDs; (**d**) Comparison of high-resolution C 1s XPS spectra of the y-CDs and r-CDs.

**Figure 5 nanomaterials-09-00529-f005:**
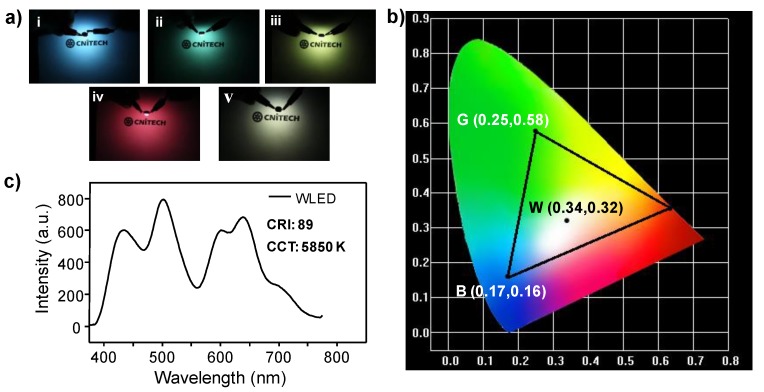
(**a**) Photographs of the monochromic (i–iv) and white (v) LEDs fabricated by encapsulation of b-CDs@ polyvinylpyrrolidone (PVP) (i), g-CDs@PVP (ii), y-CDs@PVP (iii), r-CDs@PVP (iv), and b-, g- and r-CDs@PVP mixtures (v) on UV LED chips (365 nm). (**b**) Commission Internationale de L’Eclairage (CIE) color coordinates of monochromic red, green, blue, and white LEDs. (**c**) Emission spectra and lighting performance of the white LED.

**Table 1 nanomaterials-09-00529-t001:** The quantitative fitting results of C1s XPS for the CDs.

Sample	C 1s			
	C–C/C=C (%)	C–N (%)	C–O (%)	COOH (%)
b-CDs	84.38	12.25	3.37	-
g-CDs	49.33	17.67	26.66	7.34
y-CDs	75.67	18.30	6.03	-
r-CDs	43.24	11.43	34.96	10.37

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
