# Peer review of "Preparation of Multicolor Photoluminescent Carbon Dots by Tuning Surface States"

_nanomaterials, 2019, doi:10.3390/nano9040529_

Reviewer 1 Report

This paper describes the straightforward preparation of red carbon dots with relatively high quantum yield and the preparation of mixtures of blue, green and red CD resulting on white emitters useful for RGB LED preparation. This is a scientifically interesting work well organized and well written.

Author Response

Reviewer #1

This paper describes the straight forward preparation of red carbon dots with relatively high quantum yield and the preparation of mixtures of blue, green and red CD resulting on white emitters useful for RGB LED preparation. This is a scientifically interesting work well organized and well written.

Response: Thanks these positive comments from the reviewer to our work.

Reviewer 2 Report

In this report, Prof. Hengwei Lin and co-workers showed the syntheses and properties of emissive carbon dots (CDs) by tuning the surface state.  They synthesized CDs in the presence of tartaric acid (TA).  And the resultant CDs exhibited multicolor photoluminescence (PL) available for white LEDs.  This manuscript is recommended for acceptance with some small modifications below:

1. Although the authors mentioned PL quantum yield, the measurement and calculation procedures are not clear for the readers.  And is the PL quantum yield value absolute or relative?  Furthermore, for example, “….. a high PL quantum yield up to be 22.01% ……(Page 1, Line 25)” in the Abstract is very strange.  By considering the significant figure, the value should be expressed as “22.0%”.  The authors should revise them in the next manuscript.

 2. Page 2, Line 76; I think that the authors mention “In this study,…” instead of “Recently,…”.

 3. Page 4, Line 151; Why did the PL spectrum of r-CDs show multi PL peaks?  The authors should discuss it. 

 4. The authors should express clearly the function and effect of “tartaric acid” in the Abstract.

 5. I think that the English language needs major corrections for the readers.  Checking of the English language should be undertaken by negative speakers. 

 [Minor issues]

1. The authors should exchange “SI (Supporting Informations?)” by “Supplementary Materials”.

2. Captions of Figure 3c and 3g: The authors provide one AFM image in Figure 3c and 3g.  Therefore, the authors should exchange “AFM images” by “AFM image”.

3. Page 6, Lines 209 and 210; The authors should exchange “cm-1” by “cm^-1”.  “-1” are superscripts.

Author Response

 Reviewer #2

In this report, Prof. Hengwei Lin and co-workers showed the syntheses and properties of emissive carbon dots (CDs) by tuning the surface state. They synthesized CDs in the presence of tartaric acid (TA). And the resultant CDs exhibited multicolor photoluminescence (PL) available for white LEDs. This manuscript is recommended for acceptance with some small modifications below:

1. Although the authors mentioned PL quantum yield, the measurement and calculation procedures are not clear for the readers. And is the PL quantum yield value absolute or relative? Furthermore, for example, “….. a high PL quantum yield up to be 22.01% …(Page 1, Line 25)” in the Abstract is very strange. By considering the significant figure, the value should be expressed as “22.0%”. The authors should revise them in the next manuscript.

Response: As the reviewer’s suggestion, the measurement procedure for quantum yields had been added in the revised manuscript (page 3, Line 130-134). In addition, all the recorded significant figure of QYs values had been corrected according to the reviewer’s suggestion.

2. Page 2, Line 76; I think that the authors mention “In this study,…” instead of “Recently,…”

Response: The corresponding correction had been made in the revised manuscript.

 3. Page 4, Line 151; Why did the PL spectrum of r-CDs show multi PL peaks? The authors should discuss it. 

Response: Generally, a multi-peak PL feature could be observed from an organic fluorophore that containing conjugated aromatic π systems due to the presence of distinct degenerate energy levels. As we discussed in this manuscript, the r-CDs were proposed to form via condensation and polymerization reactions between tartaric acid (TA) and o-PDs (producing aromatic π systems contained sub-fluorophores), and then further carbonization under high temperature and high pressure conditions. Thus, the multi peaks PL feature may be attributed to the uncarbonized sub-fluorophores that covered on the surface of the r-CDs. In fact, a similar multi-peak PL feature of CDs had been reported previously (Small 2018, 14, 1703919).

 4. The authors should express clearly the function and effect of “tartaric acid” in the Abstract.

Response: Thanks the reviewer’s suggestion. The function and effect of “tartaric acid” had been added in the Abstract section of the revised manuscript (page 1, line 22).

 5. I think that the English language needs major corrections for the readers. Checking of the English language should be undertaken by negative speakers.

Response: We had carefully checked and corrected the language for this manuscript again. In addition, we had also asked colleagues who have more experience in writing English papers to help us correcting this manuscript. We believe the language of this manuscript is acceptable for publication at this stage.

[Minor issues]

1. The authors should exchange “SI (Supporting Informations?)” by “Supplementary Materials”.

Response: Done.

2. Captions of Figure 3c and 3g: The authors provide one AFM image in Figure 3c and 3g.  Therefore, the authors should exchange “AFM images” by “AFM image”.

Response: Sorry for this mistake and which had been corrected in the revised manuscript.

3. Page 6, Lines 209 and 210; The authors should exchange “cm-1” by “cm^-1”.  “-1” are superscripts.

Response: Thanks the reviewer. Such corrections had been made in the revised manuscript.

Reviewer 3 Report

Attached PDF

Author Response

Reviewer #3

Jinag et al. reported the synthesis and characterization of multicolor photoluminescence carbon dots (CDs) by tuning surface states of these dots with addition of tartaric acid (TA) into m-phenylenediamine (mPD) and o-phenylenediamine (oPD) solutions. Based on the morphology and structural analyses, authors pointed out that increased surface oxidation and carboxylation would be responsible for the observed significant red-shifts in emissions of CDs and could be employed to make tunable PL CDs. Most importantly, the synthesized red-emissive CDs are found to have high PL quantum yield (~22.01%) and covering the spectral region from red to NIR showing the great potentials in optoelectronic devices and bioimaging. Authors diligently also prepared multicolor phosphors by mixing corresponding CDs (blue, green and red) with polyvinylpyrrolidone (PVP) to fabricate multicolor and white light emitting diodes (LEDs). The fabricated white LED was found to be one of the best performances of white LEDs based on CDs. Undoubtedly; these CDs with high emission quantum efficiencies would certainly have potential applications in sophisticated optoelectronic devices and bioimaging. I strongly recommend the publication of this research article in Nanomaterials, journal of international repute; however authors need to address the following comments:

Questions and revision suggestions:

1. In the Figure 1(d), the emission has two very broad peaks in the range of 400 to 500 nm, unlike authors mentioned that b-CDs, g-CDs, and y-CDs show unfeatured single peak emission. Authors need to mention it clearly in the text and write the probable mechanism for the same.

Response: It’s known from the literatures that CDs usually exhibit broad FL emission spectra due to the presence of multiple emissive origins, including surface state, core state and defect state, etc. However, according to the time-resolved PL spectra of b-CDs, g-CDs, and y-CDs (Figure S3), all of these CDs exhibit mono-exponential decay, indicating only one radiation transition dominating their emission process. Strictly speaking, the PL emission spectra of b-CDs, g-CDs and y-CDs comprise multiple emissions but with one of them dominating, and apparently showing a broad feature. To avoid a possible misunderstanding, this relevant description had been modified in the revised manuscript (page 4, line 154).

In addition, I am sorry that I did not observe the reviewer mentioned: “In the Figure 1(d), the emission has two very broad peaks in the range of 400 to 500 nm”.

2. In the Figure S1(a), the emission at 320 nm excitation is higher than 340 nm excitation and then emission intensity increases with excitation from 365 nm to 460 nm and then suddenly decreased at 480 nm excitation. On the contrary, in Figure S1 (b), emission increases with increase of excitation wavelength (365-540 nm). Authors need to address this issue. Needless to say that English language must be carefully revised.

Response: It’s common that the FL emission intensity of a phosphor directly depends on its excitation (spectrum) properties, of which being determined by its chemical structure. So, we did not see any problems of the results shown in Figure S1 and did not understand why the reviewer has such a concern.

    In addition, to the problem of English language, please see our response to the reviewer 2, question 5.

3. I would recommend authors to mention in the future scope of this work that these CDs could also be used in liquid crystals (LCs) to tune their photonic/display properties and vice-versa and may cite the following papers: (i) P. Goel et al. Liquid Crystals 39, 927-932 (2012), (ii) G. Singh et al. Liquid Crystals 39, 185-190 (2012), (iii) G. Singh et al. Liquid Crystals 44, 444-452 (2017). I think that these CDs would also have great potentials for electrically/thermally switchable LC-based photoluminescent devices.

Response: The reviewer pointed out a possible application field of CDs in the future (i.e., liquid crystals based photoluminescent devices), but this current study is mainly about development of a new method for the preparation of multicolor FL CDs by tuning their surface state. So, we don’t think it’s necessary and relevant to cite those literatures that the reviewer mentioned.

Round  2

Reviewer 2 Report

The authors have done nice revisions.

Reviewer 3 Report

Attached
